# Exosomes Derived from Human Amniotic Fluid Mesenchymal Stem Cells Preserve Microglia and Neuron Cells from Aβ

**DOI:** 10.3390/ijms23094967

**Published:** 2022-04-29

**Authors:** Manuela Zavatti, Martina Gatti, Francesca Beretti, Carla Palumbo, Tullia Maraldi

**Affiliations:** Department of Biomedical, Metabolic and Neural Sciences, University of Modena and Reggio Emilia, 41125 Modena, Italy; manuela.zavatti@unimore.it (M.Z.); martina.gatti@unimore.it (M.G.); francesca.beretti@unimore.it (F.B.); carla.palumbo@unimore.it (C.P.)

**Keywords:** neuroinflammation, Alzheimer, stem cells, exosomes

## Abstract

Background: Neuroinflammation is involved in neuronal cell death that occurs in neurodegenerative diseases such as Alzheimer’s disease (AD). Microglia play important roles in regulating the brain amyloid beta (Aβ) levels, so immunomodulatory properties exerted by mesenchymal stem cells may be exploited to treat this pathology. The evidence suggests that the mechanism of action of human amniotic fluid stem cells (hAFSCs) is through their secretome, which includes exosomes (exo). Methods: We examined the effect of exosomes derived from human amniotic fluid stem cells (hAFSCs-exo) on activated BV-2 microglia cells by lipopolysaccharide (LPS) as a neuroinflammation model. To investigate the exo effect on the interplay between AD neurons and microglia, SH-SY5Y neuroblastoma cells treated with Aβ were exposed to a conditioned medium (CM) obtained from activated BV-2 or co-culture systems. Results: We found that the upregulation of the markers of pro-inflammatory microglia was prevented when exposed to hAFSC-exo whereas the markers of the anti-inflammatory macrophage phenotype were not affected. Interestingly, the hAFSC-exo pretreatment significantly inhibited the oxidative stress rise and apoptosis occurring in the neurons in presence of both microglia and Aβ. Conclusion: We demonstrated that hAFSC-exo mitigated an inflammatory injury caused by microglia and significantly recovered the neurotoxicity, suggesting that hAFSC-exo may be a potential therapeutic agent for inflammation-related neurological conditions, including AD.

## 1. Introduction

Alzheimer’s disease (AD) is the main cause of dementia and affects about 5.7 million Americans and over 30 million people worldwide [1]. It is characterized by the degeneration and irreversible loss of neurons and synapses in the brain, impairing memory, personality and cognitive functions. The extracellular deposition of Aβ protein, a 40–42 amino acid peptide generated by proteolytic cleavages of the amyloid precursor protein (APP), and intraneuronal neurofibrillary tangles (NFTs) of hyperphosphorylated tau protein are linked to the pathogenesis of AD [2,3]. Oxidative stress is closely related to AD because it induces Aβ aggregation and facilitates the phosphorylation and polymerization of tau; in turn, the protein deposition exacerbates oxidative stress, generating a vicious cycle [4].

A critical ROS source in the AD brain are activated microglia that rapidly respond to Aβ deposition and neuron damage, interacting with astrocytes and neurons, assuming phagocytic phenotypes, participating in Aβ clearance and releasing pro-inflammatory cytokines, nitric oxide and reactive oxygen species (ROS) [5,6]. Therefore, microglia counteract Aβ deposition [7], but high levels of Aβ trigger a prolonged inflammatory condition that leads to an overproduction of inflammatory mediators and a high level of nitric oxide that, in the presence of ROS, generates ONOO^−^ and other reactive nitrogen species (NO) that are crucial contributors to oxidative stress in AD [8]. This demonstrates a neurotoxic action ascribed to the M1 phenotype whereas the neuroprotective M2 microglial phenotype is associated with the generation of anti-inflammatory components such as IL-10 and IL-4 [9].

The search for treatments capable of targeting the early pathological changes of Alzheimer’s disease (AD) such as oxidative stress and neuroinflammation is an important challenge.

A beneficial effect of MSCs on neurodegeneration has been reported, demonstrating an inhibitory effect in vitro of hBM-MSCs on oxidative stress and pro-inflammatory mediators. Specifically, human bone marrow-derived mesenchymal stem cells (hBM-MSCs) significantly inhibited the lipopolysaccharide (LPS) activation of microglial cells [10]. The observation that an MSC treatment has the potential to ameliorate Aβ pathology, modulating oxidative stress through the regulation of microglial functions, was recently confirmed by Yokokawa et al. [11] both in vitro, by co-culturing the mouse microglial cell line MG6 with MSCs, and in vivo, using an AD mouse model. MSCs induced the switch of the microglial phenotype from M1 to M2 and reduced the secretion of pro-inflammatory cytokines. Moreover, an MSC transplantation enhanced microglia accumulation around Aβ deposits, triggering microglial Aβ uptake and elimination. Furthermore, MSCs injected into the tail vein ameliorated spatial memory through a lower Aβ deposition in the cortex and hippocampus.

These results suggest that the neuronal therapeutic effect of stem cells is at least, in part, due to the maintenance of a redox homeostasis [12] that preserves neuronal functions in addition to the release of essential neurotrophic factors. The secretome of MSCs—namely, the soluble part (such as neurotrophic factors) or extracellular vesicles (EVs) (such as exosomes (exo))—is able to preserve hippocampal neurons from oxidative stress and avoid synapse damage prompted by amyloid β oligomers. This effect on neurons was demonstrated with the use of secretome derived from human Wharton’s jelly mesenchymal stem cells [13] or even from human amniotic fluid stem cells (hAFSCs) in a recent study of our group [14].

These observations highlight that the interplay between microglia and neuron cells, which generates a dangerous vicious cycle, could be a beneficial target to better interfere with the AD pathogenesis. Therefore, a co-culture of microglial cells and neurons was predicted be a good in vitro model to investigate the efficacy of EVs derived from MSCs in the interplay occurring during the AD pathogenesis. We exposed SH-SY5Y to Aβ protein as an AD model in the presence of activated microglia cells (BV-2). The aim of this work was to dissect the therapeutic role of hAFSC-exo singularly in both cell types and in a model more similar to an in vivo scenario by using transwell co-culture systems (Graphical Abstract; image elaborated with BioRender).

## 2. Results

### 2.1. hAFSC Exosome Characterization

The presence of typical exosome markers such as CD81, CD63 and CD9 after the isolation process from all three samples of hAFSC-exo was compared with the one of the hAFSC total lysate with an ELISA test. All the markers were more expressed in the exo samples obtained from different donors (Figure 1A) except for CD63, but this was in accordance with the fact that CD63 is expressed by MSCs [15]. Interestingly, an immunomodulating molecule particularly involved in microglia regulation—namely, TGFβ1 [9]—was expressed into the exo, as previously demonstrated [16].

A transmission electron microscopy (TEM) analysis was performed in order to measure the dimensions of the isolated vesicles. Figure 1B shows the representative images of one sample of exosomes isolated from hAFSC-CM. The diameter of the vesicles was around 70 nm. The images of Figure 1C show the nanoparticle tracking analysis (NTA) performed on hAFSC-exo. The median diameter of the particles was 125 nm, compatible with the characteristics of the exosomes. The number of vesicles obtained from the CM derived from 1 × 10^6^ hAFSCs was around 1 × 10^9^. The vesicles derived from the three AFSC donors were pooled and were rather homogenous, at least in terms of the dimensions and exosome marker positivity, as shown in Figure 1A.

### 2.2. Anti-Inflammatory Activity

The in vitro anti-inflammatory activity of the isolated exo was investigated in microglial BV-2 cells activated by lipopolysaccharide (LPS), the most widely used inflammatory mediator to activate microglial cells in vitro and trigger the pro-inflammatory signaling cascade [17]. We analyzed the expression of the typical markers of the activated microglia (CD86, IL1R1) to test the effect of the exo on the macrophagic switch between the M1 and the M2 phenotypes. The exo treatment induced a decrease in the M1 marker expression (Figure 2). As reactive molecules—including NO, as a product of inducible nitric oxide synthase (iNOS)—are crucial mediators of neuroinflammation, we then measured the effects of the exo on this inflammatory mediator in the LPS-activated BV-2 microglial cells. The production of NO diminished in the same cells when pretreated with the exo (Figure 2C). On the other hand, all the markers of neuroprotective microglia such as arginase1 and TGFβ were not affected (Figure 2A,B).

### 2.3. Does Exo Have a Protective Effect on the Co-Culture Systems?

At first, we analyzed the effect of the isolated exo on SH-SY5Y exposed to Aβ in the presence of a CM obtained from microglial BV-2 cells activated by lipopolysaccharide (LPS), as shown in Figure 3. 

Interestingly, the hAFSC-exo pretreatment significantly increased the SH-SY5Y cell viability (Figure 4A) by inhibiting the oxidative stress induced by the microglial CM and Aβ (Figure 4B). The intracellular ROS content was measured as a ratio between the probe fluorescence and MTT absorbance because the cell viability was differently modulated in each condition. The ROS increase was likely linked to an activation of the apoptotic pathway; caspase7 and PARP cleavages were induced by a BV-2-derived CM whereas the exo pretreatment restrained these effects (Figure 4C).

In order to test the regenerative effect of hAFSC-exo on neurons, the changes in cell morphology were investigated. Figure 5A shows that the neuron processes—stained with typical markers of mature neurons, βtubulin III and MAP2—were affected by Aβ and/or the CM treatments, as expected. Notably, the exo presence was able to reduce these events, such as decreasing the length and thickness of neurites, and even partially restore the expression of MAP2 (Figure 5B).

We then assessed the effect of hAFSC-exo on a more dynamic system, a transwell co-culture between SH-SY5Y on the bottom side and activated BV-2 on the top (see Figure 3). The ROS level and cell viability were tested on neuron cells. As expected, the treatments with Aβ and LPS-activated BV-2, alone or together, caused an increase in ROS content that was linked to a decrease in cell viability. Notably, the co-culture with non-activated BV-2 was considered to be the real control point because the presence of microglial cells induced an increase in SH-SY5Y viability (Figure 6A). However, an exo exposure counteracted the ROS rise, thus avoiding cell death (Figure 6A,B), confirming the results obtained from the BV-2 CM exposure (Figure 4A,B).

The analysis of the amyloid β peptide 1–42 (Aβ42) content in the supernatants of the co-culture revealed that the highest value was obtained in the case of the co-treatment (BV-2 + exogenous Aβ), which was as expected, but also the treatment with BV-2 alone induced a clear Aβ detection. The presence of exo dramatically reduced the presence of Aβ (Figure 6C), suggesting an increase in protein accumulation clearance.

## 3. Discussion

Alzheimer’s disease (AD) is a progressive neurodegenerative brain disorder associated with a loss of memory and cognitive function. Amyloid beta (Aβ) aggregates in particular are known to be highly neurotoxic and lead to neurodegeneration. In such cases, microglia play important roles in regulating the brain Aβ levels. Therefore, the blockade or reduction of Aβ aggregation is a promising therapeutic approach in AD.

Although previous studies have described the potential of MSC on microglia or neurons in experimental models of AD [18,19], here, we demonstrate the therapeutic efficacy of extracellular vesicles (EVs) secreted by perinatal stem cells, hAFSCs, on an in vitro AD model, including human neurons and microglia in a co-culture.

Perinatal cells, including cells from the umbilical cord and amniotic fluid, display intrinsic immunological properties that very likely inhibit the activation, and modulate the functions, of various inflammatory cells of the innate and adaptive immune systems [20]. These immunological properties, along with their easy availability and lack of ethical concerns, make perinatal cells very useful/promising in regenerative medicine. In recent years, EVs have gained great interest as a new therapeutic tool in regenerative medicine, being a cell-free product potentially capable—thanks to the growth factors, miRNA and other bioactive molecules they convey—of modulating the inflammatory microenvironment, thus favoring tissue regeneration. Several studies have reported that EVs from perinatal tissues are comparable with the parental cells when transplanted in several preclinical models of inflammatory-mediated diseases such as neurodegenerative diseases [21,22,23]. Notably, amniotic fluid stem cells are the youngest of them because they are the only ones obtained in the second trimester. Furthermore, EVs have the advantage of being a cell-free therapy and, therefore, have reduced risks compared with those associated with the transplantation of live cells.

Here, we characterized EVs obtained by the ultracentrifugation of hAFSC-CM through TEM and NTA analyses in order to check the number of isolated particles and the dimensions from both techniques. These analyses confirmed that we collected vesicles compatible with the features of exosomes because exosomal vesicles typically have a divot in their center, unlike microvesicles and apoptotic bodies. Consistently, the typical markers of exosomes such as CD9, CD81 and CD63 were detected in all exo samples, suggesting an enrichment of these vesicles in our suspension.

Regarding AD, it has been reported that human umbilical cord blood-derived mesenchymal stem cells (hUCB-MSCs) can increase, through a paracrine action, the ability of microglial cells to clear Aβ [24] through an overexpression of the Aβ-degrading enzyme neprilysin in microglia [25]. When hUCB-MSCs were transplanted into a transgenic mouse model of Alzheimer’s disease, Aβ42 plaques in the hippocampus and other regions were decreased by the active migration of hUCB-MSCs toward Aβ deposits. On the other side, we have previously reported an inhibitory effect of hAFSC-EV on Aβ aggregation in primary neurons obtained from a transgenic AD mouse model [14]. Moreover, we have demonstrated the presence of TGFβ into the exo [16]; here, we again checked these data because our previous studies were conducted with EVs isolated by a commercial kit, but also because this factor plays a pivotal role in regulating microglia effects [9]. We previously performed proteomic and Western blot analyses of EVs isolated by a commercial kit. In addition to TGFβ, HGF, pentraxin and IDO, other immunomodulating factors, were detected. Moreover, we demonstrated that SOD1, an antioxidant enzyme, was expressed into these vesicles [16].

Here we examined the effect on SH-SY5Y neuroblastoma cells grown with a conditioned medium from BV-2 microglia cells exposed to hAFSC-exo that had been activated by lipopolysaccharide (LPS) as a neuroinflammation model. We found that the exo pretreatment on the activated BV-2 cells significantly increased SH-SY5Y cell viability by inhibiting the apoptosis normally induced by an activated microglia-conditioned medium. The exo were able to inhibit the activity of the neuroinflammatory microglial cells, decreasing the release of reactive oxygen/nitrogen species in BV-2 cells stimulated with LPS. Thus, hAFSC-exo had protective effects against Aβ-induced indirect neurotoxicity, either by inhibiting the production of NO whilst favoring TGFβ and arginase1 in BV-2 cells or by protecting SH-SY5Y cells against these inflammatory mediators. The fact that microglial cells could receive TGFβ from hAFSC-exo could generate a positive loop because TGFβ1 and TGFβ signaling are indispensable for microglia maturation, adult microglia homeostasis and the control of microglia activation in central nervous system pathologies [26].

A transwell co-culture was established to determine the direct effect of microglia treated with hAFSC-exo on the survival of SH-SY5Y cells exposed to Aβ. The cell viability of neuroblastoma cells increased when co-cultured with BV-2 cells, but the presence of LPS stimulated BV-2-induced cell death, which was even more evident with the co-treatment Aβ + activated BV-2, as expected. The neuron ROS levels were higher in the presence of these treatments. Interestingly, the exo exposure prevented these effects on ROS and viability in a significant manner for the worse condition; namely, the co-treatment Aβ + activated BV-2. This efficacy could be due to a combined effect on microglial cells; the inhibition of the M1 phenotype sustained the M2 one and favored Aβ clearance, as demonstrated by the ELISA test. A speculation on the mechanism underpinned to this effect could rely on our previous observation on the protein content of AFSC-EVs [16]; apolipoprotein E (ApoE) was present with a high score and it is a ligand for the activation receptor expressed on myeloid 2 cells (TREM2). TREM2 can act as a multifaceted actor in microglial functions in the brain homeostasis of AD, not only by influencing the microglial functions in amyloid and tau diseases, but also by participating in inflammatory responses, acting alone or with other molecules such as ApoE as a ligand [27]. It was demonstrated that myeloid cells can clear Aβ directly through the TREM2-mediated uptake of lipoprotein-Aβ complexes, modeling ApoE–Aβ interactions observed in vivo [28]. Moreover, human genetic data indicate that microglial dysfunction contributes to AD, as exemplified by the identification of variants encoding TREM2 [29]. Therefore, the modulatory effect of hAFSC-exo on microglial cells could also be due to the presence of ApoE into the vesicles.

Collectively, our results indicated that hAFSC-exo effectively mitigated an inflammatory injury caused by an LPS-conditioned medium from microglia, possibly due to reductions in the iNOS activity and the release of resolving factors and provide evidence of a neuroprotective role of these vesicles through the blocking of negative consequences of microglial activation. Furthermore, exo protected neuron cells from Aβ aggregate-induced toxicity by reducing the aggregation of Aβ and significantly reduced the rise in oxidative stress stimulated by the interplay between microglia and the neuron cells (Graphical Abstract).

These findings support a role of hAFSC-exo as a cell-free therapy for the treatment of neuroinflammatory and degenerative diseases, including AD, in which more than one cell target should be considered to obtain relevant outcomes.

## 4. Materials and Methods

### 4.1. Amniotic Fluid Stem Cell Isolation

The hAFSCs were obtained from amniotic fluids collected from 3 healthy pregnant women at the 16th week of gestation who underwent an amniocentesis at maternal request (not for fetal anomalies) at the Unit of Obstetrics and Gynecology, Policlinico Hospital of Modena (Italy). The amniocentesis was performed under continuous ultrasound guidance in a sterile field with 23-gauge needles. The risks related to the procedure and the purpose of the study were explained to all patients before the invasive procedure and the ob-gyn specialist collected a signed consent before starting the exam (protocol 360/2017, dated 15 December 2017 and approved by Area Vasta Emilia Nord). For this study, supernumerary (unused) flasks of AF cells cultured in the Laboratory of Genetics of the TEST Lab (Modena, Italy) for 2 weeks were used.

The hAFSCs were isolated as previously described [30]. Human amniocentesis cultures were harvested by trypsinization and subjected to c-kit immunoselection by MACS technology (Miltenyi Biotec, Bergisch Gladbach, Germany). The hAFSCs were routinely subcultured at a 1:3 dilution and not allowed to grow beyond a 70% confluence in the culture medium (αMEM) supplemented with 20% fetal bovine serum (FBS), 2 mM L-glutamine, 100 U/mL penicillin and 100 μg/mL streptomycin (all from EuroClone Spa, Milano, Italy).

### 4.2. Exosome Isolation

The hAFSCs were grown in 75 cm^2^ flasks until subconfluence (around 1 × 10^6^ cells). Before the extracellular vesicle extraction, the cells were maintained for 4 days in 10 mL of a culture medium deprived of FBS in order to exclude contamination by extracellular vesicles included in the FBS solution. The secreted part of the conditioned medium (CM) was centrifuged at 300 × *g* for 10 min at 4 °C and then concentrated up to 2 mL by using centrifugal filter units with a 3 K cutoff [31]. The supernatant was again centrifuged at 10,000× *g* for 30 min at 4 °C and the supernatant was transferred to poly(propylene) ultracentrifuge tubes (13.2 mL, Beckman Coulter). The supernatant was then centrifuged at 100,000× *g* for 90 min at 4 °C in a Beckman Coulter Optima L-90 K centrifuge with a SW-41 rotor; the supernatant was discarded and the pellets were resuspended in 13 mL PBS and centrifuged again at 100,000× *g* for 90 min at 4 °C. The pellet was resuspended into 100 μL PBS for the spectrophotometric and NTA analyses.

After dilution at 1:1000, the size distribution and concentration of the exosomes were analyzed by a nanoparticle tracking analysis using a ZetaView particle tracker from ParticleMetrix (Inning am Ammersee, Germany).

### 4.3. Electron Microscopy

Transmission electron microscopy (TEM) assays were performed as previously reported [16]. Briefly, exosome pellets from the cells or sera were suspended in and fixed with 4% paraformaldehyde and 4% glutaraldehyde in 0.1 M of a phosphate buffer (pH 7.4) at an incubation temperature and kept at 4 °C until the analysis. A drop of each exosome sample was placed on a carbon-coated copper grid and immersed in 2% phosphotungstic acid solution (pH 7.0) for 30 s. The preparations were examined under TEM (JEM-1200EX, JEOL Ltd., Japan) at an acceleration voltage of 80 kV.

### 4.4. Cell Culture and Treatments

The SH-SY5Y cell line was purchased from Sigma-Aldrich (ECACC 94030304) (St. Louis, MO, USA) and was grown in high-glucose DMEM supplemented with 10% (*v*/*v*) FBS, 2 mM L-glutamine, 50 U/mL penicillin and 50 μg/mL streptomycin, as previously reported [14]. The cells were used for experiments after inducing their differentiation with all-trans retinoic acid (10 μM) for 7 days. Differentiated SH-SY5Y cells were treated for 24 h with 10 µM Aβ 1–42 fibrils (GenScript, Piscataway, NJ, USA) obtained after incubation at 37° C for 24 h.

BV-2 murine microglial cells were a kind gift of Prof. Elisabetta Blasi (University of Modena and Reggio Emilia, Modena, Italy). The cells were cultured in RPMI supplemented with 10% (*v*/*v*) low-endotoxin FBS (Euroclone, Milano, Italy), 2 mM L-glutamine, 50 U/mL penicillin and 50 μg/mL streptomycin. The cells were maintained in a humidified incubator at 37 °C with 5% CO_2_ and detached by vigorous shaking. The BV-2 cells were pretreated with 0.5 × 10^9^ exosomes/10^6^ cells for 4 h before the addition of 1 ug/mL LPS for 24 h.

For experiments with the BV-2-conditioned medium (CM), SH-SY5Y cells were plated (4 × 10^5^ cells per well) on 6-well plates. After the differentiation procedure described above, the medium was replaced with a CM for 24 h.

Transwells are a well-characterized 3D model of a co-culture and have been used extensively for neuronal co-cultures including SH-SY5Y and BV-2. BV-2 cells were plated (3 × 10^5^ cells per well) on top of the transwell inserts in an RPMI medium w/o FBS. After the treatments with exosomes and LPS as previously described, the transwell inserts (BioBasic Inc., Toronto, Canada) were placed on top of the 24-well containing SH-SY5Y (4 × 10^5^ cells per well) in a medium containing 50% RPMI and 50% DMEM. See Figure 3 for a diagram of the treatment. The cells were treated with 10 µM Aβ 1–42 for 24 h.

### 4.5. MTT Assay

The cells (BV-2 and SH-SY5Y) were seeded in 96-well plates in 100 μL of the culture medium with 4 replicates for each condition at a density of 10,000 cells/well. At the end of the treatments, 0.5 mg/mL MTT was added and incubated for 3 h at 37 °C. After incubation, the medium was removed and acidified isopropanol was added to solubilize the formazan salts [32]. The absorbance was measured at 570 nm using a microplate spectrophotometer (Appliskan, Thermo-Fisher Scientific, Vantaa, Finland).

### 4.6. ROS and NO Detection

To evaluate the intracellular ROS levels, a dichlorodihydrofluorescein diacetate (DCFH-DA) assay was performed similar to as previously described [33]. Cells (BV-2 and SH-SY5Y) were seeded in 96-well plates with 5 replicates for each condition at a density of 10,000 cells/well. After the cell treatments, the cell culture medium was removed and 5 μM DCFH-DA was incubated in PBS for 30 min at 37 °C and 5% CO_2_. The cell culture plate was washed with PBS and the fluorescence of the cells was read at 485 nm (excitation) and 535 nm (emission) using the multiwall reader Appliskan (Thermo-Fisher Scientific, Vantaa, Finland). The cellular autofluorescence was subtracted as a background using the values of the wells not incubated with the probe.

A Griess assay (Sigma Aldrich, Milan, Italy) was performed to detect the NO production. The supernatants and a Griess reagent were mixed at a ratio of 1:1 and after 10 min the samples were analyzed by a spectrophotometer at 546 nm. The NO production was expressed in uM.

### 4.7. Cellular Extract Preparation

Total cell lysates (TL) were obtained as previously described [34]. Briefly, the cells were treated with a lysis buffer (20 mM Tris-Cl, pH 7.0; 1% Nonidet P-40; 150 mM NaCl; 10% glycerol; 10 mM EDTA; 20 mM NaF; 5 mM sodium pyrophosphate; and 1 mM Na_3_VO_4_) and freshly added to a protease inhibitor cocktail (Sigma Aldrich) and para-nitrophenylphosphate (Sigma Aldrich) at 4 °C for 20 min. The lysates were sonicated, cleared by centrifugation and immediately boiled in SDS, reducing the sample buffer.

### 4.8. SDS PAGE and Western Blot

The TL from BV-2 and SH-SY5Y cells were processed as previously described [35]. Primary antibodies were raised against the following molecules: actin (diluted 1:5000) (Sigma-Aldrich, St Louis, MO, USA); PARP; arginase1; caspase 7; IL-RL1 (Santa Cruz Biotechnology, CA, USA); TGFβ; CD86 (Novus Biologicals, Milano, Italy); and βtubulin III (Cell Signaling Technology, Lieden, The Netherlands) (diluted 1:1000).

Secondary antibodies, used at 1:3000 dilutions, were all from Thermo Fisher Scientific (Waltham, MA, USA).

### 4.9. ELISA Assays

CD9, CD63, CD81 and TGF β were quantified in the lysed exo or in the hAFSC cell extract (TL). Briefly, the lysed exo was obtained by treating the exo with a lysis buffer at a ratio of 1:3 (*v*/*v*) followed by 3 cycles of freeze and thaw. A total of 1 μg of total protein extract from the exo or cell lysate (TL) was quantified by an ELISA kit (Cusabio Technology, Houston, TX, USA) according to the manufacturer’s protocol.

Amyloid beta peptide 1–42 (Aβ42) was analyzed in the supernatants of SH-SY5Y. Media of neurons and BV-2 in transwell co-culture assays, exposed or not to the exo, were quantified by BETA-APP42 according to the manufacturer’s protocol of the ELISA kit (Aviva Systems Biology, San Diego, CA, USA).

### 4.10. Immunofluorescence and Confocal Microscopy

For the immunofluorescence analysis, SH-SY5Y cells seeded onto coated coverslips were processed and confocal imaging was performed using a Nikon A1 confocal laser scanning microscope, as previously described [36].

Primary antibodies to detect MAP2 (dil 1:100) and βtubulin III (dil 1:500) (Cell Signaling Technology, Lieden, Netherlands) were used following the datasheet-recommended dilutions. Alexa secondary antibodies (Thermo Fisher Scientific, Waltham, MA, USA) were used at 1:200 dilutions.

The confocal serial sections were processed with ImageJ software to obtain three-dimensional projections. The image rendering was performed by Adobe Photoshop software.

### 4.11. Cellular Morphology

The cellular images were acquired using an EVOS XL Core Cell Imaging System (Thermo Fisher Scientific, Vantaa, Finland) as previously reported [14]. The parameter and area were measured with ImageJ. The cellular elongation was calculated using the following formula:Cellular elongation = p2/4π ∗ A
where p is the cellular perimeter, π is equivalent to 3.14 and A represents the cellular area.

### 4.12. Statistical Analysis

The experiments were performed in triplicate (biological replicates). For quantitative comparisons, the values were reported as the mean ± SD based on a triplicate analysis for each sample. To test the significance of the observed differences amongst the study groups, a one-way ANOVA with a Bonferroni post hoc test or a Student’s *t*-test were applied. A *p*-value < 0.05 was considered to be statistically significant. The statistical analysis and plot layout were obtained by using GraphPad Prism^®^ release 6.0 software.

## Figures and Tables

**Figure 1 ijms-23-04967-f001:**
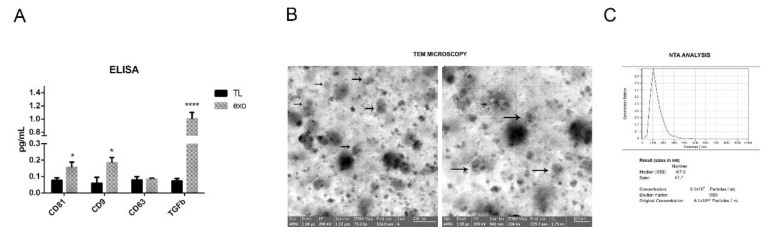
Exosome characteristics. (**A**) ELISA analysis of exo markers and TGFβ1 on the three lysates of isolated exosomes and relative hAFSCs (TL: total lysate). * *p*-value < 0.05 (CD81 *p* = 0.041; CD9 *p* = 0.039); **** *p*-value < 0.0001 for TGFβ1. (**B**) Representative images of transmission electron microscopy (TEM) shown at different magnifications. Arrows indicate vesicles. (**C**) Representative nanoparticle tracking analysis (NTA) performed on exosome suspension with ZetaView.

**Figure 2 ijms-23-04967-f002:**
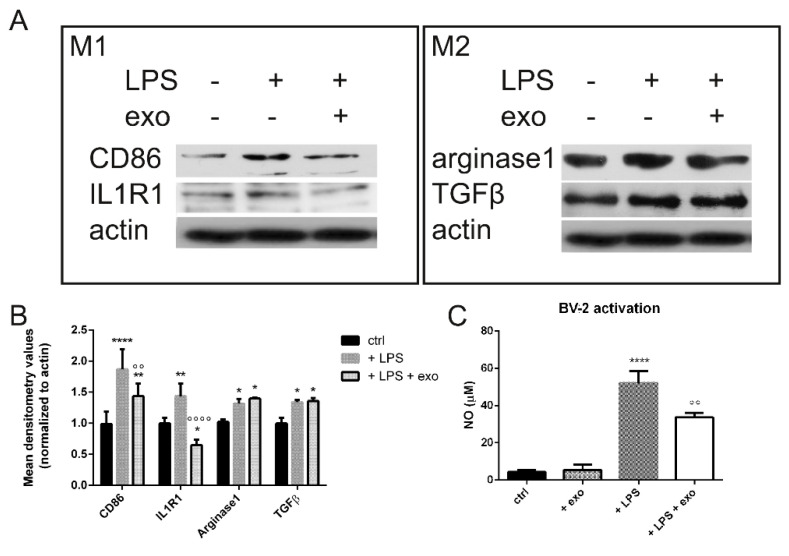
M1 and M2 microglia switch. (**A**) WB analysis of markers of M1 and M2 phenotypes in presence or absence of LPS and exo (24 h). (**B**) The graph represents the mean ± SD of densitometric analysis of 3 experiments, normalized to actin values. CD86: **** *p*-value < 0.0001; ** *p* = 0.003; °° *p* = 0.004. IL1R1: * *p*-value = 0.021; ** *p* = 0.004; °°°° *p*-value < 0.0001. Arginase1: + LPS * *p* = 0.046; + LPS + exo * *p* = 0.013. TGFβ: + LPS * *p* = 0.024; + LPS + exo * *p* = 0.018. (**C**) Graph showing changes in NO production. °° *p*-value = 0.001; **** *p*-value < 0.0001.

**Figure 3 ijms-23-04967-f003:**
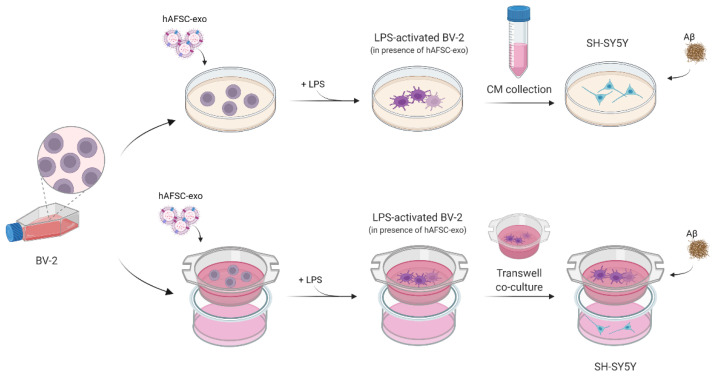
Diagram of experimental design. SH-SY5Y exposed to BV-2 CM or to BV-2 co-culture in presence or absence of Aβ and exo. Image elaborated with BioRender.

**Figure 4 ijms-23-04967-f004:**
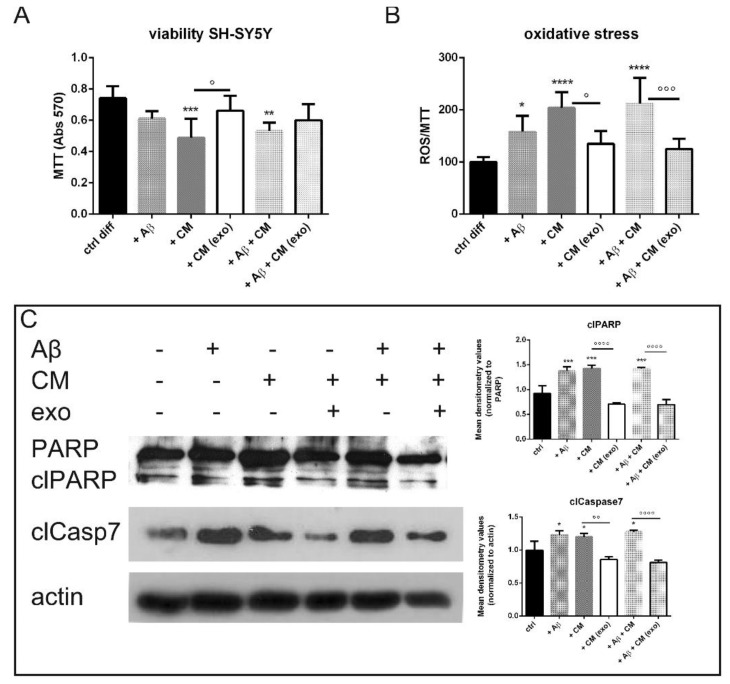
Indirect effect of hAFSC-exo on SH-SY5Y viability. (**A**) Graph of MTT assay of neuron cells exposed to BV-2 CM and/or Aβ in presence or absence of exo. ° *p* = 0.024; ** *p* = 0.009; *** *p* = 0.002. (**B**) Graph of ROS content normalized to MTT in neuron cells exposed to BV-2 CM and/or Aβ in presence or absence of exo. * *p* = 0.027; ° *p* = 0.011; °°° *p* = 0.0004; **** *p*-value < 0.0001. (**C**) Representative WB experiment performed on SH-SY5Y exposed to BV-2 CM pretreated or not with exo. The graphs represent the mean ± SD of densitometric analysis of 3 experiments normalized to actin values. PARP: Aβ *** *p* = 0.0003; CM *** *p* = 0.0001; Aβ + CM *** *p* = 0.0001; °°°° *p* < 0.0001. Caspase7: Aβ * *p* = 0.014; CM * *p* = 0.035; Aβ + CM * *p* = 0.012; °° *p* = 0.0011; °°°° *p* < 0.0001.

**Figure 5 ijms-23-04967-f005:**
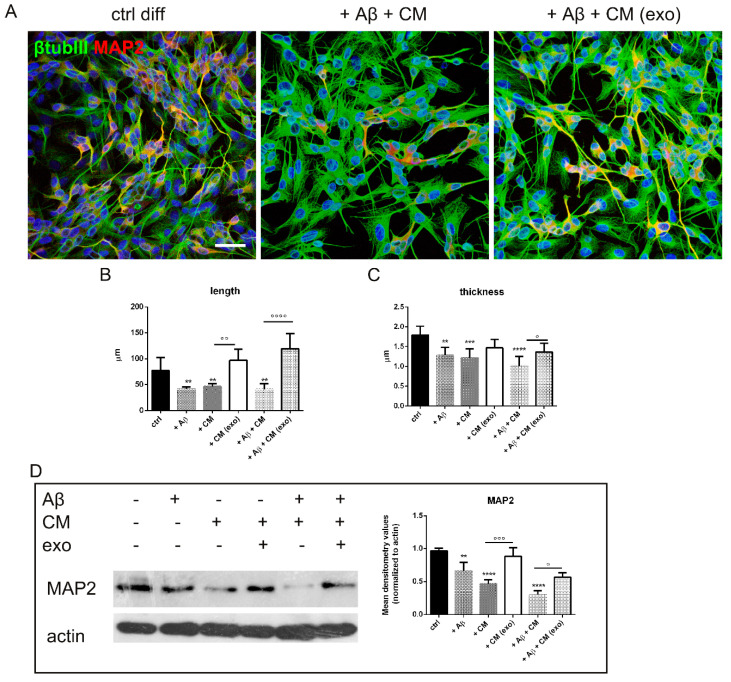
Indirect effect of hAFSC-exo on SH-SY5Y morphology. (**A**) Representative images with DAPI (blue), βtubulin III (green) and MAP2 (red) signals of neurons treated as previously reported. Scale bar: 20 µm. (**B**,**C**) Graphs showing length and thickness values measured in 100 SH-SY5Y cells per condition considering βtubulin III staining. Length: Aβ ** *p* = 0.008; CM ** *p* = 0.008; Aβ + CM ** *p* = 0.009; °° *p* = 0.002; °°°° *p*-value < 0.0001. Thickness: ° *p* = 0.035; ** *p* = 0.0042; °° *p*-value = 0.0029; *** *p* = 0.0006; **** *p*-value < 0.0001. (**D**) Western blot analysis of total lysate of SH-SY5Y cells treated or not with BV-2 CM and/or Aβ in presence or absence of exo, then revealed with anti-MAP2 and anti-actin as a loading control. The graph represents the mean ± SD of densitometric analysis of 3 experiments normalized to actin values. ° *p* = 0.025; ** *p* = 0.005; °°° *p* = 0.0003; **** *p*-value < 0.0001.

**Figure 6 ijms-23-04967-f006:**
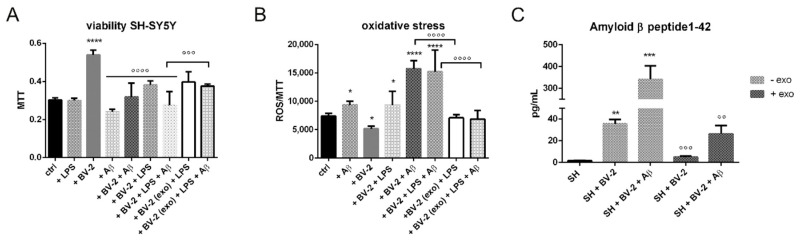
The effect of hAFSC-exo on transwell co-culture SH-SY5Y BV-2. (**A**) Graph of MTT assay of neuron cells exposed to (+/- LPS) BV-2 and/or Aβ in presence or absence of exo. * *p* = 0.048; ****, °°°° *p*-values < 0.0001. (**B**) Graph of ROS content (normalized to MTT) in neuron cells exposed to (+/- LPS) BV-2 and/or Aβ in presence or absence of exo. Aβ * *p* = 0.049; BV-2 * *p* = 0.042; BV-2 + LPS * *p* = 0.040; ****, °°°° *p*-values < 0.0001. (**C**) ELISA test of Aβ concentration in conditioned medium obtained from the co-culture of neurons and microglia cells treated as previously reported. ** *p* = 0.001; °° *p* = 0.002; *** *p* = 0.0002; °°° *p* = 0.0003.

## Data Availability

Not applicable.

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
