# Peer review of "Exosomes Derived from Human Amniotic Fluid Mesenchymal Stem Cells Preserve Microglia and Neuron Cells from Aβ"

_ijms, 2022, doi:10.3390/ijms23094967_

Round 1
Reviewer 1 Report
Research Article: “Exosomes derived from human amniotic fluid mesenchymal stem cells preserve microglia and neuron cells from Aβ.”
In this manuscript authors have investigated the the effect of exosomes derived from human amniotic fluid stem cells (hAFSC-exo) on activated BV-2 microglia cells and the exo effect on the interplay between AD-neurons and microglia. They uncovered that upregulation of markers of pro-inflammatory microglia was prevented when exposed to hAFSC-exo and hAFSC-exo pretreatment significantly inhibited the oxidative stress rise and apoptosis occurring in neurons in presence of both microglia and Aβ. This study highlighted that hAFSC-exo mitigated inflammatory injury caused by microglia, and significantly recovered the neurotoxicity. This is an interesting study and this reviewer has few comments that can improve the quality of the manuscript.
1, Figure 1C quality may be improved (high resolution).
2, At least one illustrative figure may be provided as to highlight the summary of this study.
3, Did authors characterize the exosomes and find what they are carrying? Explain in discussion.
Author Response
Dear Editor,
We thank you and the reviewers for the opportunity to improve the paper. We answered to all the criticisms underlined by the reviewers. We hope that the manuscript is now suitable for publication in IJMS.
Reviewer 1
We really thank the reviewer for the comments and suggestions to our study. Here below the answers to the comments.
1, Figure 1C quality may be improved (high resolution).
We agree with the reviewer. The image has been replaced.
2, At least one illustrative figure may be provided as to highlight the summary of this study.
Following the reviewer suggestion, we added a graphical abstract.
3, Did authors characterize the exosomes and find what they are carrying? Explain in discussion.
Actually we previously performed proteomic analysis, therefore we added a sentence in the discussion. “ Indeed, we previously performed proteomic and western blot analyses of EV isolated by a commercial kit: beside TGFβ, even HGF, pentraxin and IDO, other immunomodulating factors, were detected. Moreover, we demonstrated that SOD1, an antioxidant enzyme, was expressed into these vesicles [15].”
Reviewer 2 Report
The manuscript entitled “Exososomes derived from human amniotic fluid mesenchymal stem cells preserve microglia and neuron cells from Ab” describes the antinflammatory effects of hAFSC exo in an in vitro model of neuro inflammation mimicked by using activated Bv-2 microglial cell line and the “neuron” differentiated SH-SY5Y cell line. Bv-2ells were activated through the classical LPS inflammatory insult. Moreover authors used Ab 1-42 fibrils treatment to model one of the hall marks of the AD pathology.
The authors have already used the use of hAFSC exo as therapeutic approach in other pathological contexts, actually there refer to these studies trough all the manuscript since some methologial approaches presented has been already describes in previous published studies.
Overall the study has been well designed and methodologically performed, but there are two major points that should be clarified and some parts that should be impaired from my point of view.
Major
- Have authors used the exo obtained from the three different subjects AFSC? The results obtained whith each of the hAFSC exo preparation should be included in this study.
- Authors describe a decrease in the surnatant Ab 1-42 levels in the presence of exo, thus suggesting an increase in this protein accumulation clearance. Did authors measured the intracellular levels of Ab 1-42? How authors explain this decrease?
Minor
- Pg 2, authors argue that “the neuronal therapeutic effect of stem cells is at least in part due to the maintenance of an appropriate redox state that preserves neuronal function”. What “appropriate redox state” means?
- Abbreviations should be specified through all the text, also in figure captions. Moreover the description of the abbreviation should appear the first time it is used, for instance NTA, even though there is a short list of Abbreviations at the end of the manuscript.
- P values from statistical studies should be specified in the text, results section or captions.
- The image of figure 3 could be much improved.
- Figure 4, graph A, Y axes represents absorbance values from MTT assay, so the title of Y axes should be absorbance and the wavelength at which the MTT assay has been read. The same in Figure 6, graph A.
- Figure 4, graph B, Y axes, what is the meaning of ROS/ MTT?, What has been represented in the graph? The same in Figure 6, graph B.
- Figure 4, graph D includes the results of DNA damage obtained through immunocytochemistry after the staining with the pH2A antibody. Representative ICC images have not been included, is there any reason?
- Figure 5, how authors have measured the length and thickness values included in graphs B and C, respectively?, did they used a software validated to perform these analyses? What antibody stained cells were used to perform these analyses?
- How many biological and technical replicates have been used in every assay performed? In some figures’ captions (Fig 2, 4, 5) the number of replicates is included (n=3) but it is not clear if they are biological or technical replicates. This information should be included in the Material and Method section or in every assay performed.
- Are cellular extracts different from whole lysates? What type of samples are used to perform the SDS PAGE and western blot?
- The dilutions of the antibodies used for western blot and ICC should be specified.
- Pg 6, last line, “… since they are the only ones… “
- Pg 7, third paragraph, there is an extra “that”.
- Pg 9, first paragraph, last line, LPS concentration is 1 microg/mL, there is a typo “-1”.
- Pg 10, third paragraph, line 3, did authors mean7 “… was quantified by BETA-AMYLOID 42… “
- Pg 9, third paragraph, second line, “… including SH-SY5Y… “.
Author Response
Dear Editor,
We thank you and the reviewers for the opportunity to improve the paper. We answered to all the criticisms underlined by the reviewers. We hope that the manuscript is now suitable for publication in IJMS.
Reviewer 2
We really thank the reviewer for the comments and suggestions to our study. Here below the answers to the comments.
Major
- Have authors used the exo obtained from the three different subjects AFSC? The results obtained whith each of the hAFSC exo preparation should be included in this study.
We agree with the reviewer comment, this point needs to be clarified. We added a sentence in the first part of Results in order to specify how the experiments have been performed after the vesicle isolation and characterization part. In brief, we decided to pool together the vesicle suspensions. “Vesicles deriving from the three AFSC donors have been pooled, being rather homogenous at least in term of dimensions and exosome marker positivity, as shown in Figure 1 A.”
- Authors describe a decrease in the surnatant Ab 1-42 levels in the presence of exo, thus suggesting an increase in this protein accumulation clearance. Did authors measured the intracellular levels of Ab 1-42? How authors explain this decrease?
We thank the reviewer for this comment, since we agree that this point is interesting to investigate in a further study. Actually, we have some hypothesis based on our previous observations. Therefore, we added a paragraph in the discussion part, being aware that this is only a speculation. “A speculation on the mechanism underpinned to this effect could relies on our previous observation on the protein content of AFSC-EVs [15]: apolipoprotein E (ApoE) was present with a high score and is a ligand for activation receptor expressed on myeloid 2 cells (TREM2). Indeed, TREM2 can act as a multifaceted actor in microglial functions in the brain homeostasis of AD, not only by influencing microglial functions in amyloid and tau diseases, but also by participating in inflammatory responses, acting alone or with other molecules, such as (ApoE) as a ligand [Qin et al., 2021]. It was further demonstrated that myeloid cells can clear Aβ directly through TREM2- mediated uptake of lipoprotein-Aβ complexes, modeling the ApoE-Aβ interactions observed in vivo [Yeh et al., 2016]. Moreover, human genetic data indicate that microglial dysfunction contributes to AD, as exemplified by the identification of variants encoding TREM2 [Andreone et al., 2020]. Therefore the modulatory effect of hAFSC-exo on microglial cells could also be due to the presence of ApoE into the vesicles.”
Minor
- Pg 2, authors argue that “the neuronal therapeutic effect of stem cells is at least in part due to the maintenance of an appropriate redox state that preserves neuronal function”. What “appropriate redox state” means?
We agree with the reviewer that this expression could be confusing, therefore we replaced it with clearer one, based on studies of Prof. Ursini: redox homeostasis (from “Redox homeostasis: The Golden Mean of healthy living” Ursini et al., 2016 Redox Biol).
- Abbreviations should be specified through all the text, also in figure captions. Moreover the description of the abbreviation should appear the first time it is used, for instance NTA, even though there is a short list of Abbreviations at the end of the manuscript.
We apologize for this lack of clarity. We added the description of TEM and NTA in the text and into the legend.
- P values from statistical studies should be specified in the text, results section or captions.
We apologize again for this lack of clarity. We added the p values were needed.
- The image of figure 3 could be much improved.
We agree with the reviewer. The image has been replaced.
- Figure 4, graph A, Y axes represents absorbance values from MTT assay, so the title of Y axes should be absorbance and the wavelength at which the MTT assay has been read. The same in Figure 6, graph A.
We agree with the reviewer. The wavelength has been added in the two graphs.
- Figure 4, graph B, Y axes, what is the meaning of ROS/ MTT?, What has been represented in the graph? The same in Figure 6, graph B.
ROS fluorescence values depend on the cell number, therefore, in case of cell death, as after Aβ exposure etc., these values need to be normalized to the cell number (MTT value). In order to clarify this point, we added a sentence into the text. “Intracellular ROS content was measured as a ratio between the probe fluorescence and MTT absorbance, since cell viability was differently modulated in each condition.”
- Figure 4, graph D includes the results of DNA damage obtained through immunocytochemistry after the staining with the pH2A antibody. Representative ICC images have not been included, is there any reason?
We agree with the reviewer. The most representative images have been added in Figure 4D.
- Figure 5, how authors have measured the length and thickness values included in graphs B and C, respectively?, did they used a software validated to perform these analyses? What antibody stained cells were used to perform these analyses?
We apologize again for this lack of clarity. We added the method already used and described in our paper Gatti et al. [13].
“Cellular images were acquired using EVOS XL Core Cell Imaging System (Thermo Fisher Scientific, Vantaa, Finland). Parameter and area were measured with ImageJ. Cellular elongation was calculated using the following formula:
Cellular elongation= p2/4π*A
where p is the cellular perimeter, π is equivalent to 3,14 and A represents the cellular area.”
- How many biological and technical replicates have been used in every assay performed? In some figures’ captions (Fig 2, 4, 5) the number of replicates is included (n=3) but it is not clear if they are biological or technical replicates. This information should be included in the Material and Method section or in every assay performed.
As reported in M&M section, every ROS, NO and MTT assays was performed with 5 or 4 technical replicates, but each experiment has been performed 3 times, so as biological replicates. For this reason, in captions related to WB analysis or IF experiment the number of replicates was already included (n=3). Moreover, in section 4.11, the paragraph describing statistical analysis, the number of replicates were already cited. However, we added in brackets (biological replicates) for clarity, as suggested by the reviewer.
- Are cellular extracts different from whole lysates? What type of samples are used to perform the SDS PAGE and western blot?
Total lysates, cellular extracts and whole lysates have been used indifferently. For clarity, we replaced all with Total lysates.
- The dilutions of the antibodies used for western blot and ICC should be specified.
We apologize again for this lack of clarity. We added the dilutions.
- Pg 6, last line, “… since they are the only ones… “
We thank the reviewer for this note. We corrected the sentence.
- Pg 7, third paragraph, there is an extra “that”.
We thank the reviewer for this note. We corrected the sentence.
- Pg 9, first paragraph, last line, LPS concentration is 1 microg/mL, there is a typo “-1”.
We thank the reviewer for this note. We corrected the sentence.
- Pg 10, third paragraph, line 3, did authors mean7 “… was quantified by BETA-AMYLOID 42… “
Actually the name of the ELISA test is BETA-APP and the code is OKEH00815.
- Pg 9, third paragraph, second line, “… including SH-SY5Y… “.
We thank the reviewer for this note. We corrected the sentence.
Round 2
Reviewer 2 Report
- “A speculation on the mechanism underpinned to this effect could “rely” on our previous observation on the protein content of AFSC-EVs [15]: apolipoprotein E (ApoE) was present with a high score and “it” is a ligand for activation receptor expressed on myeloid 2 cells (TREM2). Indeed, TREM2 can act as a multifaceted actor in microglial functions in the brain homeostasis of AD, not only by influencing microglial functions in amyloid and tau diseases, but also by participating in inflammatory responses, acting alone or with other molecules, such as (ApoE) as a ligand [Qin et al., 2021]. It was further demonstrated that myeloid cells can clear Aβ directly through TREM2- mediated uptake of lipoprotein-Aβ complexes, modeling the ApoE-Aβ interactions observed in vivo [Yeh et al., 2016]. Moreover, human genetic data indicate that microglial dysfunction contributes to AD, as exemplified by the identification of variants encoding TREM2 [Andreone et al., 2020]. Therefore the modulatory effect of hAFSC-exo on microglial cells could also be due to the presence of ApoE into the vesicles.”
- Pg 13, first paragraph, line 3, please, specify what are you meaning with the word “compounds”.
- The reference (from “Redox homeostasis: The Golden Mean of healthy living” Ursini et al., 2016 Redox Biol) should be included in the text’s manuscript.
- P values from statistical studies should be specified in the text, results section or captions.
We apologize again for this lack of clarity. We added the p values were needed.
When performing statistical analysis, there is the possibility of obtaining the value of p. This is the value needs to be specified in the manuscript, not the relationship between number of asterisks and p interval values.
- The image of figure 3 could be much improved.
We agree with the reviewer. The image has been replaced.
The meaning of this criticism was to improve the figure in the content, not in the magnification. For instance, Bv-2 cells from flask to petri dish are activated, LPS activated? In the presence / absence of exo? The suggestion is to include all these conditions in the figure so that they can be understood at first sight.
- Figure 4, graph D includes the results of DNA damage obtained through immunocytochemistry after the staining with the pH2A antibody. Representative ICC images have not been included, is there any reason?
We agree with the reviewer. The most representative images have been added in Figure 4D.
It is hard to see pH2A positive cells (green) in the images. The green spots seem artifacts instead of cell DNA. On the other hand the % of DNA-damaged cells has been included in the graph. In the “worse” condition (Ab+CM) only 4-5 cells are DNA-damaged. How can authors make any conclusion with this so low number of damaged cells? Moreover the error bars are huge for such low numbers. This result is not credible. Authors should use a different technique to study DNA damage.
Author Response
We really thank the reviewer for the comments and suggestions to our study. Here below the answers to the comments.
- “A speculation on the mechanism underpinned to this effect could “rely” on our previous observation on the protein content of AFSC-EVs [15]: apolipoprotein E (ApoE) was present with a high score and “it” is a ligand for activation receptor expressed on myeloid 2 cells (TREM2). Indeed, TREM2 can act as a multifaceted actor in microglial functions in the brain homeostasis of AD, not only by influencing microglial functions in amyloid and tau diseases, but also by participating in inflammatory responses, acting alone or with other molecules, such as (ApoE) as a ligand [Qin et al., 2021]. It was further demonstrated that myeloid cells can clear Aβ directly through TREM2- mediated uptake of lipoprotein-Aβ complexes, modeling the ApoE-Aβ interactions observed in vivo [Yeh et al., 2016]. Moreover, human genetic data indicate that microglial dysfunction contributes to AD, as exemplified by the identification of variants encoding TREM2 [Andreone et al., 2020]. Therefore the modulatory effect of hAFSC-exo on microglial cells could also be due to the presence of ApoE into the vesicles.”
We thank the reviewer for this note. We corrected the sentence.
- Pg 13, first paragraph, line 3, please, specify what are you meaning with the word “compounds”.
We thank the reviewer for underlining us this mistake. We corrected the sentence.
- The reference (from “Redox homeostasis: The Golden Mean of healthy living” Ursini et al., 2016 Redox Biol) should be included in the text’s manuscript.
Ursini et al is now reference n 12.
- When performing statistical analysis, there is the possibility of obtaining the value of p. This is the value needs to be specified in the manuscript, not the relationship between number of asterisks and p interval values.
We apologize for the misunderstanding. We added all the p values in the legends.
- The meaning of this criticism was to improve the figure in the content, not in the magnification. For instance, Bv-2 cells from flask to petri dish are activated, LPS activated? In the presence / absence of exo? The suggestion is to include all these conditions in the figure so that they can be understood at first sight.
We apologize for the misunderstanding. Figure 3 has been changed in order to make the experimental design clear at first sight.
- Figure 4, it is hard to see pH2A positive cells (green) in the images. The green spots seem artifacts instead of cell DNA. On the other hand the % of DNA-damaged cells has been included in the graph. In the “worse” condition (Ab+CM) only 4-5 cells are DNA-damaged. How can authors make any conclusion with this so low number of damaged cells? Moreover the error bars are huge for such low numbers. This result is not credible. Authors should use a different technique to study DNA damage.
We agree with the reviewer that it is hard to see green signal (pH2A) when DAPI is superimposed. The highly fluorescent spots are obviously artefacts, but nuclei stained in green can be easily noticed in images of separated signals, as here below (in pdf file). However, the scepticism of the reviewer on this data drives us to remove the images and the relative graph (Figure 4D), since this data was not essential for the study so that it was not even commented in the discussion part.

Round 3
Reviewer 2 Report
Authors have answered to all the criticisms, the new version of the manuscript is, in my opinion, accepted for pubblication.